# Concept Understanding in Large Language Models: An Empirical Study

**Jiayi Liao***
University of Science and Technology of China
ljy0ustc@mail.ustc.edu.cn

**Xu Chen**[†] **& Lun Du**
Microsoft Research Asia
{xu.chen, lun.du}@microsoft.com

## Abstract

Large Language Models (LLMs) have demonstrated their superior comprehension and expressiveness across a wide range of tasks, and exhibited remarkable capabilities in real-world applications. Hence, it is crucial to investigate their potential and limitations for trustworthy performance in both academia and industry. In this paper, we focus on exploring LLMs' ability to understand concepts, especially abstract and concrete ones. To this end, we construct a WordNet-based dataset containing a subset for abstract concepts and a subset for concrete concepts. We select six pre-trained LLMs and conduct a classic NLP task, hypernym discovery, as evidence of LLMs' comprehension ability in understanding concepts. The experimental results suggest that the LLM's understanding of abstract concepts is significantly weaker than that of concrete concepts.

## 1 Introduction

In the past few years, Large Language Models (LLMs) have become the frontiers of academic research. The recently released ChatGPT has further exhibited the potential of LLMs in various downstream tasks with advanced technologies and engineering efforts (Brown et al., 2020). On the other hand, the language understanding ability of LLMs also affects their deployments in real-world scenarios, making exploration of this research problem a new topic.

This paper focuses on the understanding ability of LLMs from the perspective of abstract and concrete concepts. Such explorations are inspired by the fact that different real-world tasks usually require the understanding ability on different levels of abstraction. A better comprehension of concrete concepts may be desirable in tasks related to physical entities, like classifying the type of animals. In contrast, other tasks ask the LLMs to know more about abstract entities, such as distinguishing different human emotions or logical reasoning.

To achieve this goal, we first construct a new dataset D-Concept based on WordNet, a lexical database of semantic relations between words (Miller, 1995). The dataset follows the setting of a classic NLP task of hypernym discovery, and it can reflect the language model's ability to understand abstract and concrete concepts. Nouns in WordNet are explicitly divided into an abstract branch and a physical branch, and thus it is an appropriate prior. Therefore, we follow this division to construct two subsets based on the abstract branch and the physical branch, respectively. Each data sample consists of a pair of entities from the corresponding branches. The hypernym discovery task is to determine whether these two entities are hypernyms or not. The explored LLMs range from BERT to the GPT series models (including OpenAI text embedding model and ChatGPT). Experimental results show that, in the hypernym discovery task, the LLM's performance on abstract concepts is worse than on concrete concepts, indicating the improvement of room for LLMs.

Our contributions can be summarized as follows: (1) We construct a new dataset for the hypernym discovery task to compare the LLM's understanding ability on abstract and concrete concepts. (2) We investigate the performance of LLMs at different scales on this task and discover that the performance improves as the model scales up while consistently worse on abstract concepts than concrete ones.

---

*Work performed during the internship at MSRA.
[†]Corresponding author.

## 2 EXPERIMENT

### 2.1 EXPERIMENT SETUP

**Dataset.** WordNet groups nouns into sets of cognitive synonyms (synsets), each expressing a distinct concept and can be regarded as an entity. Abstract branch and physical branch are two main components in WordNet, representing abstract concepts (like "Fairness" and "Happiness") and concrete concepts (like "Animal" and "furniture"), respectively. To explore how LLMs represent concepts, we visualize embeddings of concepts in WordNet generated by GPT embedding model (text-similarity-ada-001) with T-SNE. As shown in Figure 1, embeddings of two types of concepts are roughly divided into two clusters, revealing that there indeed exist some differences between them.

To dive deeper into such differences, we create a new dataset for the concept understanding task, i.e., hypernym discovery. Hypernymy relation (e.g., "bed"-"furniture" where furniture is a hypernymy and bed is a hyponym) is one typical lexical relation on WordNet due to the nature that WordNet is a hierarchical graph with entities as nodes and hypernymy relations as edges. When constructing the dataset, on each branch, we first randomly choose a distance value $d$ for positive examples (i.e., one entity and its corresponding hypernym, as WordNet is a tree-like structure and ancestors in different heights of a single hyponym are all its hypernyms). Secondly, we randomly select the first entity, and then the second entity is sampled from an entity set where each entity is at a distance $d$ from the first entity. Negative entity pairs are randomly and separately chosen. There are 10,000 samples in total, split as Train: Valid: Test = 2: 4: 4.

**Settings.** We compare 6 LLMs, including BERT (Devlin et al., 2018), T5 (Raffel et al., 2020), CLIP (Radford et al., 2021), OpenClip (Cherti et al., 2022), GPT embedding model (Brown et al., 2020) and ChatGPT. We freeze their embeddings and add a linear layer for binary classification. Cross-Entropy Loss is used as the loss function, and Accuracy (ACC), AUC and F1 score are evaluation metrics. For ChatGPT, we use the prompt-answering paradigm to obtain its answers (i.e., we use "*Is {entity1} the hypernym of {entity2}?*" as a prompt to get "*yes*" or "*no*" from ChatGPT).

### 2.2 EXPERIMENTAL RESULT

Table 1 shows LLMs' performance on different types of concepts under the hypernymy discovery task. We can find that LLMs consistently perform worse for abstract concepts compared with concrete concepts. It implies that there is improvement room for LLMs in learning high-quality embeddings, especially for abstract concepts. The poor results on abstract concepts may raise the risk of handling abstract concept-related tasks when applying LLMs. We can also discover that when the model scale increases, the performance on both tasks also improves except for ChatGPT, where the embedding can not be obtained and is tested without fine-tuning or few-shot learning.

Table 1: Hypernym discovery result.

| Model | Abstract Concept | | | Concrete Concept | | |
|---|---|---|---|---|---|---|
| | ACC | AUC | F1 | ACC | AUC | F1 |
| BERT | 85.28 | 85.28 | 84.35 | 89.65 | 89.61 | 89.12 |
| T5 | 85.85 | 85.85 | 85.43 | 89.45 | 89.41 | 88.88 |
| CLIP | 87.93 | 87.93 | 87.62 | 91.53 | 91.49 | 91.13 |
| OpenClip | 88.85 | 88.85 | 88.47 | 91.60 | 91.56 | 91.22 |
| GPT Embedding | 89.45 | 89.45 | 89.04 | 91.98 | 91.94 | 91.60 |
| ChatGPT | 64.13 | 64.14 | 44.10 | 78.98 | 78.75 | 73.04 |

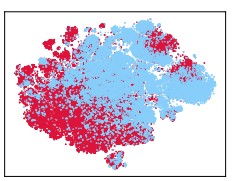

Figure 1: Visualization for abstract concepts (red) and concrete concepts (blue).

## 3 CONCLUSION

We construct a new WordNet-based dataset for the hypernym discovery task in order to explore the understanding ability of large language models on abstract and concrete concepts. Experimental results show that LLMs struggle with abstract concepts, regardless of their model size, which inspires researchers to mitigate this gap in the future.

## URM STATEMENT

Author Jiayi Liao meets the URM criteria of ICLR 2023 Tiny Papers Track.

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

## A  D-CONCEPT DATASET BRIEFS

D-Concept dataset includes 21962 synsets in total, consisting of their name and definition from the WordNet. The synset pairs in D-Concept are separated into two subsets, one for abstract concepts and the other for concrete concepts. An example of hypernym pairs from abstract concept subset is "bushel"-"volume unit" with the distance $d = 3$ where bushel means *a British imperial capacity measure (liquid or dry) equal to 4 pecks* and volume unit means *a unit of measurement of volume or capacity*. An example of hypernym pairs from concrete concept subset is "*Ayrshire*"-"*Cattle*" with distance $d = 2$, where Ayrshire denotes *hardy breed of dairy cattle from Ayr, Scotland* and Cattle is *domesticated bovine animals as a group regardless of sex or age*.

The number of synset pairs is 2000, 4000, and 4000 for training, validating, and testing, respectively. Detailed information about the dataset is illustrated in Figure 2 and Figure 3.

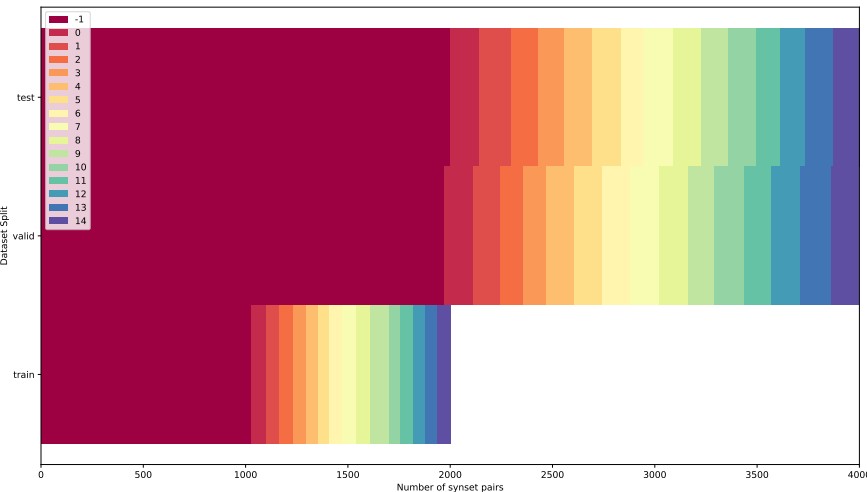

Figure 2: The dataset split of abstract concepts and the distribution of path distances on WordNet tree between abstract synset pairs. -1 represents no hypernym relationship between the synset pair.

## B  EXPERIMENTAL DETAILS

**Backbones.**  (1) BERT (Devlin et al., 2018): We take the pooler outputs of the bert-base-uncased model as embeddings, with a dimension of 768. (2) T5 (Raffel et al., 2020): We take the average pooling of the last hidden layer of the google/t5-v1_1-large model as embeddings, with a dimension of 1024. (3) CLIP (Radford et al., 2021) and OpenClip (Cherti et al., 2022): The text embedding dimensions we get from text encoders of CLIP and OpenClip are 768 and 1024, respectively. (4) GPT series model (Brown et al., 2020): We adopt OpenAI's embeddings service (text-similarity-ada-001) and OpenAI's chat completion service. The embedding dimension of the text-similarity-ada-001 model is 1024. And the prompt for ChatGPT is "*Is {synset-1} a hypernym of {synset-2}? {synset-1} means {the definition of synset-1}. {synset-2} means {the definition of synset-2}. Please directly answer YES or NO. (Do not return any explanation or any additional information.)*". Providing ChatGPT with the meaning of synset aims to help with entity disambiguation, and formatting the answer from ChatGPT can facilitate post-processing for the evaluation.

**Training Process.**  The learning rate will reduce to 1/3 of its original value if the loss does not decrease for 20 epochs. Validating is conducted every 10 epochs and the best models are chosen based on AUC. Early stopping is adopted if the AUC on the validation set has not increased for 100 epochs. The learning rate of MLP's parameters is log uniform searched in the range of [0.0001, 0.1]. Each result presented in Table 1 is the best of 40 repeated experiments.

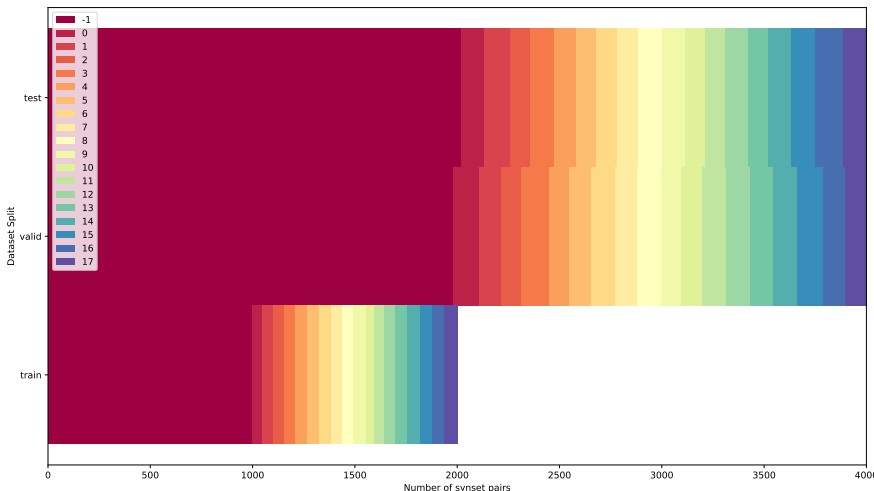

Figure 3: The dataset split of physical concepts and the distribution of path distances on WordNet tree between physical synset pairs. -1 represents no hypernym relationship between the synset pair.

## C  RELATED WORK

**LLMs in Concept Understanding.**  While LLMs demonstrate extraordinary abilities in various NLP tasks, it is debated heatedly whether LLMs have the capability of concept understanding, or if their strong performance simply attributes to the statistical correlations discovered as the scale of models grows larger (Mitchell & Krakauer, 2023). Works like Sahu et al. (2022) have been devoted to exploring whether LLMs can understand concepts. Our work chooses to analyze this question from the perspective of abstract concept understanding ability of LLMs.

**LLMs in Hypernym Discovery Task.**  Previous works like  Vulić et al. (2020) and  Hanna & Mareček (2021) have explored linguistic knowledge in LLMs (e.g., BERT) with tasks such as lexical relation prediction (including hypernym discovery). They indicate that LLMs still contain limited knowledge of hypernyms. Datasets for hypernym discovery, a classic NLP task, have also been proposed in many works such as Baroni & Lenci (2011), Snow et al. (2004), Roller et al. (2014), Vyas & Carpuat (2017), Camacho-Collados et al. (2018). However, our D-Concept Dataset explicitly divides concept pairs into abstract and concrete based on WordNet, and conducts the hypernym discovery task on them, respectively. This dataset formation is more convenient for research on abstract concept understanding.

