# OpenReview forum: "Concept Understanding in Large Language Models: An Empirical Study"
_ICLR.cc/2023/TinyPapers — Submitted to Tiny Papers @ ICLR 2023_

### Official Review · Reviewer_DUYr · 2023-04-01

**Confidence:** 3

**Summary Of Contributions:**

The paper explores how well Large Language Models (LLMs) understand abstract vs concrete concepts. The authors describe a new dataset created from WordNet for hypernym discovery, and then evaluate the performance for several different LLMs on both abstract and concrete concepts. Results indicate better performance for concrete concepts.

**Rating:**

Clear, Correct, and Reproducible (CCR): a submission which meets the reviewing criteria

**Strengths And Weaknesses:**

Strengths:

-	The paper clearly explains the hypernym discovery task, how it can be used to evaluate how well LLMs understand abstract vs concrete concepts, and provides clear results.

-	The authors consider a wide variety of LLM models and show consistent results across the different models.

Weaknesses:

-	The literature review is extremely limited, especially for NLP tasks related to hypernyms, despite the claim that it is “a classic NLP task”.

-	While LLMs consistently perform worse on abstract concepts, the scores are usually only a few percentage points worse. Are the differences statistically significant? Does this properly reflect the understanding for abstract concepts is “significantly weaker”?


**Suggested Changes:**

-	Provide more citations, especially around hypernym discovery.

-	Provide more analysis on the difference in results. For example, is 91.98% accuracy significantly different than 89.45% accuracy?

-	Is identifying hypernyms for abstract concepts a more difficult task for humans than identifying hypernyms for physical concepts? Providing some background literature, or even a human baseline to the results, would help contextualize the difference in results that is exhibited by LLMs

---

### Official Review · Reviewer_5SgH · 2023-04-01

**Confidence:** 4

**Summary Of Contributions:**

The paper creates a database for hypernym discovery. It also presents results on experiments illustrating the difference in performance between the understanding of abstact and concrete concepts by different LLMs

**Rating:**

Great Start (GS): a submission which meets some of the reviewing criteria but has room for improvement

**Strengths And Weaknesses:**

## Strengths
- The dataset seems to be quite a novel one in terms of examining conceptual understanding of LLMs
- It uses a large range of models, in terms of capacity

## Weaknesses
- The authors use ChatGPT which is constantly changing and therefore the results cannot be reproduced
- The difference between `abstract` and `concrete` examples isn't demonstrated. Some examples would make it a stronger stand-alone paper
- Lack of experiment details. The authors don't show any examples of the the words used for any of the results



**Suggested Changes:**

1. Provide some examples of the abstract and concrete concepts used to evaluate the models.
2. Use OpenAI's embeddings service to compare in place of ChatGPT
3. If possible, show how the results differ at different scales of `d`, the distance used

---

### Official Review · Reviewer_ixfh · 2023-04-01

**Confidence:** 3

**Summary Of Contributions:**

The paper constructs a dataset with positive and negative examples of hypernyms where the relation concept can be both abstract and physical. It then proceeds to check performance of LLMs in understanding the concept of hypernyms.

**Rating:**

Great Start (GS): a submission which meets some of the reviewing criteria but has room for improvement

**Strengths And Weaknesses:**

Strengths:

* Creates a new Wordnet-based dataset with positive and negative examples of hypernyms.
* Studies the important notion of concept-understanding capabilities of LLMs and compares multiple models with vastly different parameter size.

Improvement scopes:

* I am not convinced about the conclusion that there seem to be two clusters among the two concepts - abstract and physical. It rather seems to me from Figure 1 that there is a large overlap. Although this doesn't really impact the empirical results too much, a better discussion of this might be interesting.
* For the dataset generation, there seems to be three steps. Sampling positive examples, negative examples and pairs where words are equidistant as positive examples. I am not sure what this equidistance sampling achieves. Additionally, since the distance here is from GPT-3.5 embeddings, what kind of effect does that have on the final classification task? A discussion of this question is missing.
* For ChatGPT, multiple types of prompts could have been tried. E.g., "Can word2 be considered an example/entity of word1?". This would give a better idea both about prompt strengths and how concepts are modeled.
* Any statistics about dataset size and label distribution for the DConcept dataset is missing. A short paragraph or table would be helpful.
* For accuracy, could it be that the abstract concepts' performance suffered from having a smaller dataset? From the paper, it seems to me that both branches were separately created and trained/tested on. It would be interesting to see what would happen if training for both concepts were done together. I believe that you could still do more experiments within the current scope/range of this workshop.

**Suggested Changes:**

I would suggest to address the points made in the improvement scopes.

---

### Official Review · Reviewer_iS5E · 2023-04-01

**Confidence:** 3

**Summary Of Contributions:**

The paper investigates whether LLMs are able to comprehend concepts and develop a dataset for the task. The results show that scaling model sizes does help in better comprehension, but LLMs consistently recognize concrete concepts better as compared to abstract ones.

**Rating:**

Great Start (GS): a submission which meets some of the reviewing criteria but has room for improvement

**Strengths And Weaknesses:**

**Strengths:**

- With the explosion of applications running GPTx architectures in the backend, this paper touches on a timely issue.
- The paper introduces a new dataset for hypernym discovery, which could be a great starting point for future work.
- The authors compare multiple large models.

**Weaknesses:**

- I don't see any links to the D-Concept dataset, which is a major flaw. Any experiments to reproduce the results are severely hindered. I hope the authors add a link to it in the future.
- The conclusion proposed by the authors might be misplaced (see Suggestions).
- Missing certain relevant publications regarding similar work done in this area like:
````
Sahu, P., Cogswell, M., Gong, Y., & Divakaran, A. (2022). Unpacking Large Language Models with Conceptual Consistency. ArXiv. /abs/2209.15093
````

**Suggested Changes:**

This is an innovative paper, taking a step towards solving an issue that will be very important in the coming days.

- (From weakness)
However, my main issue with the paper is that the authors seem to be extrapolating the task of hypernym discovery/recognition to concept understanding. The success of a correct answer to the question *"Is **Word1** a hypernym of **Word2**?"* (in case of ChatGPT) may not necessarily imply a general understanding of the concepts relevant to the query.

- While this paper may serve as a great starting work, given that a majority is devoted to creating and testing LLMs on a dataset of hypernym association, the authors may want to have their abstract and introduction reflect the same. The title seems a bit misleading as well.

- Repeating a suggestion from the `Weakness` section, I urge the authors to release their developed dataset as well. It would help others in the community build upon this work.

- Given that this paper touches on a topic that anthropomorphizes LLMs, I would also suggest adding citations like [1]. I do not list this as a `Weakness` since [1] was released fairly recently. However, similar works did exist as well.

````
[1] Mitchell, M., & Krakauer, D. C. (2023). The debate over understanding in AI’s large language models. Proceedings of the National Academy of Sciences, 120(13), e2215907120.
````

---

### Meta-Review · Area_Chair_GWw5 · 2023-04-02

**Recommendation:** Invite to archive
**Confidence:** 5

**Metareview:**

My recommendation - Invite to archive

The paper constructs a dataset for the hypernyms task where the relation concept can be abstract and physical. The paper then proceeds to check the performance of LLMs in understanding the concept of hypernyms.

Given the recent advances in LLMs, this is a timely paper. The main critiques identified by the reviewers revolve around “reproducibility”. Given that the dataset is new, the authors must make it accessible. Also, an overview of the dataset will be a welcome addition to the Appendix. Furthermore, it will be very useful to have the exact settings of experiments. This again allows for the ability to reproduce (you can add it as part of the appendix). There is no page restriction on references, so include suggested literature.

Although I agree that ChatGPT changes constantly, so it is hard to reproduce. If the authors would like to include these results, clarify that there are certain restrictions around ChatGPT.

Make use of the appendix to provide clear explanations where needed.



**Summary:**

The paper constructs a dataset for the hypernyms task where the relation concept can be abstract and physical. The paper then proceeds to check the performance of LLMs in understanding the concept of hypernyms.

**Comments And Feedback To The Authors:**

Please see above. Also, please read through the reviews.

**Reason For Not Giving A Higher Recommendation:**

As mentioned above, the paper as is will not be suitable. However, if the author makes the suggested changes then it can be accepted.

**Reason For Not Giving A Lower Recommendation:**

N/A

---

### Decision · Program_Chairs · 2023-04-07

Invite to archive